

# Interventional treatment for azygos vein steal syndrome after bidirectional Glenn procedure in cyanotic congenital heart disease: a retrospective study

Zhengwei Li, Luxi Guan, Dong Luo, Meijun Liu, Haibo Hu and Xiangbin Pan

Center of Structural Heart Disease, Fuwai Hospital, National Center for Cardiovascular Diseases, Chinese Academy of Medical Sciences, Peking Union Medical College, Beijing, China

## ABSTRACT

**Objectives**. Patients with cyanotic congenital heart disease (CCHD) may continue to experience hypoxia and worsening cyanosis after undergoing a bidirectional Glenn procedure (BGP). Azygos or hemiazygos vein steal syndrome is a common cause of these complications. This study aimed to evaluate the safety and efficacy of transcatheter closure for azygos or hemiazygos vein steal syndrome following BGP in patients with CCHD.

**Methods**. A retrospective analysis was conducted on clinical data from 13 patients with CCHD who underwent transcatheter closure for azygos or hemiazygos vein steal syndrome after BGP at Fuwai Hospital between December 2007 and September 2019.

**Results**. All azygos or hemiazygos veins were successfully closed. Femoral artery oxygen saturation ($S_{FA}O_2\%$) significantly increased after closure compared to before closure ($86.94 \pm 2.63\%$ *vs* $74.98 \pm 3.53\%$, $P < 0.001$). The mean pulmonary arterial pressure (mPAP) also showed a statistically significant increase after closure but remained within normal ranges ($12.08 \pm 2.75$ mmHg *vs* $10.54 \pm 3.28$ mmHg, $P = 0.020$). The superior vena cava pressure (SVCP) showed almost no change before and after closure ($11.08 \pm 3.62$ mmHg *vs* $12.31 \pm 3.25$ mmHg, $P = 0.059$). During an average follow-up period of $25.69 \pm 9.60$ months, all patients showed significant improvement in cyanosis, and none experienced facial or neck edema. The saturation of pulse oximetry ($SpO_2$) was above 90% in all patients.

**Conclusions**. Transcatheter closure of azygos or hemiazygos vein steal syndrome in CCHD patients after BGP is a safe and effective intervention. It offers high technical success and favorable short- to mid-term outcomes, while avoiding the physical and psychological trauma associated with repeat thoracotomy.

Corresponding author
Haibo Hu,
hhb.fwhospital@hotmail.com

# INTRODUCTION

The bidirectional Glenn procedure (BGP), also known as bidirectional cavopulmonary anastomosis, involves connecting the superior vena cava to the ipsilateral pulmonary

artery to form a "cavopulmonary conduit". This surgery increases pulmonary blood flow, enhances arterial oxygen saturation, reduces cardiac load, and improves cardiac function *Calvaruso et al. (2008)*. The BGP is a palliative surgical treatment for cyanotic congenital heart disease (CCHD), generally applied in conditions characterized by underdeveloped right ventricles, such as single ventricles, tricuspid atresia, and pulmonary atresia. However, patients may still face issues of hypoxia and worsening cyanosis after BGP. Azygos or hemiazygos vein "steal" syndrome is a common cause of these problems. The azygos vein serves as an important collateral pathway connecting the superior and inferior vena cava; the hemiazygos vein often acts as a branch of the azygos vein. After BGP, due to the higher pressure in the superior vena cava (SVC) compared to the inferior vena cava (IVC), blood from the SVC can shunt through the azygos vein into the IVC, creating an extracardiac right-to-left shunt. This leads to a reduction in pulmonary blood flow and a progressive decrease in oxygen saturation. Therefore, the azygos vein is often ligated during BGP to prevent blood flow from the cavopulmonary conduit to the inferior vena cava *via* the azygos vein. However, when there is underdeveloped pulmonary artery growth or high pulmonary vascular resistance leading to higher pressures in the cavopulmonary conduit, surgeons often leave the azygos vein open to serve as a "decompression channel" for the cavopulmonary conduit. This helps prevent postoperative adverse events such as pulmonary hypertension crisis and superior vena cava obstruction syndrome. However, long-term extracardiac shunting can lead to symptoms such as chest tightness, dyspnea, and cyanosis, which significantly impair the patient's quality of life and can even be life-threatening. Such conditions require intervention to manage the azygos or hemiazygos vein. Therefore, our study included 13 cases of CCHD patients who underwent transcatheter closure of azygos or hemiazygos vein steal syndrome after BGP. The aim was to evaluate the safety and efficacy of interventional treatment and to summarize our center's experience in managing this condition, providing new insights for clinical practice in treating such diseases.

## METHODS

### Patient population

We retrospectively analyzed the clinical data of 13 CCHD patients who underwent transcatheter closure of azygos or hemiazygos vein steal syndrome after BGP at the Fuwai Hospital, Chinese Academy of Medical Sciences, between December 2007 and September 2019. Among these patients, nine were male and four were female, with ages ranging from 8 to 36 years (mean age: 13.92 ± 7.35 years), heights ranging from 112 to 172 cm (mean height: 143.92 ± 22.60 cm), and weights ranging from 21 to 52 kg (mean weight: 35.48 ± 11.34 kg). These 13 CCHD patients (including those with a single ventricle, double outlet right ventricle, tetralogy of Fallot, *etc.*) had undergone BDG surgery 2 to 22 years prior (mean: 7.62 ± 5.38 years). All patients presented with cyanosis, chest tightness and shortness of breath after exertion, palpitations, and audible heart murmur upon auscultation. In addition to these symptoms, Case 11 had hemoptysis, and Case 4 had clubbing of the fingers. 77% (10/13) of the patients were classified as New York Heart Association (NYHA) Class II, while 23% (3/13) were NYHA Class III. Before transcatheter

**Table 1  The baseline characteristics of the 13 patients.**

| Case | Sex | Age (year) | Height (cm) | Weight (kg) | BP (mmHg) | NYHA | CCHD |
|------|-----|------------|-------------|-------------|-----------|------|------|
| 1 | M | 11 | 125 | 23 | 128/96 | 2 | MID, SV, TGA, ASD, PS |
| 2 | M | 15 | 170 | 50 | 124/90 | 2 | SV, TGA, ASD, PVS, PLSVC, MID |
| 3 | M | 8 | 120 | 21 | 96/78 | 3 | DORV, VSD, PAS, TGA, PLSVC |
| 4 | M | 11 | 125 | 26 | 108/72 | 2 | SV, TTGA |
| 5 | M | 15 | 165 | 37 | 90/65 | 2 | SV, ASD, TGA, PVS, PDA, PLSVC, SAVV |
| 6 | F | 8 | 112 | 28 | 90/60 | 2 | SV, PVS, TAVSD |
| 7 | M | 9 | 113 | 26 | 85/94 | 2 | TA, VSD, ASD, PVS |
| 8 | F | 36 | 154 | 46 | 105/75 | 3 | DORV, VSD, ASD, PVS |
| 9 | M | 17 | 172 | 52 | 143/72 | 2 | DORV, ASD, VSD, PVS, TGA, AI, TS |
| 10 | F | 17 | 165 | 45 | 94/58 | 2 | DORV, ASD, VSD |
| 11 | F | 13 | 157 | 36 | 102/48 | 3 | TOF |
| 12 | M | 9 | 136 | 25 | 102/63 | 2 | TA, ASD, VSD |
| 13 | M | 12 | 157 | 47 | 113/63 | 2 | SV, TGA |

**Notes.**

M, male; F, female; BP, blood pressure; MID, mirror image dextrocardia; SV, single ventricle; TGA, transposition of the great arteries; TTGA, total transposition of the great arteries; ASD, atrial septal defect; PDA, patent ductus arteriosus; PVS, pulmonary valve stenosis; PAS, pulmonary artery stenosis; PLSVC, persistent left superior vena cava; DORV, double outlet right ventricle; SAVV, single atrioventricular valve; TAVSD, total atrioventricular septal defects; TA, tricuspid atresia; AI, aortic insufficiency; TS, tricuspid stenosis; TOF, tetralogy of fallot.

closure, all patients underwent detailed evaluations, including thorough history taking, physical examination, electrocardiography, chest X-rays, echocardiography, and cardiac catheterization with angiography. All patients except Case 5, who underwent hemiazygos vein closure, had azygos vein closure. The detailed baseline characteristics of the patients are provided in Table 1. This study was approved by the ethics committee of Fuwai Hospital (Approval NO.2021-1551), and written informed consent was obtained from all patients.

## Transcatheter closure

Each patient underwent transcatheter closure of the abnormal azygos or hemiazygos vein under local or general anesthesia, with measurements of oxygen saturation of the femoral artery ($S_{FA}O_2$%) and hemodynamic data taken both before and after the occlusion to evaluate the effectiveness of the procedure. The majority of post-occlusion data were collected immediately after the surgery or on the following day. During the operation, close monitoring of pulmonary artery pressure (PAP) and central venous pressure (CVP) is required; a trial occlusion must be performed first, ensuring that PAP and CVP do not rise significantly or remain below 16 mmHg before the release of the occluder device is considered.

(1) The patient underwent transcatheter closure of the azygos vein due to "steal" syndrome. Under local anesthesia, the operator punctured the right internal jugular vein and inserted a 5F pigtail catheter to perform angiography of the superior vena cava, pulmonary artery, and azygos vein (Fig. 1A). Through the right jugular vein, a 16-mm atrial septal defect (ASD) occluder was implanted successfully to close the proximal segment of the azygos vein. After closure, repeated angiography of the superior vena cava showed complete occlusion, with no opacification observed in the mid and distal segments of the

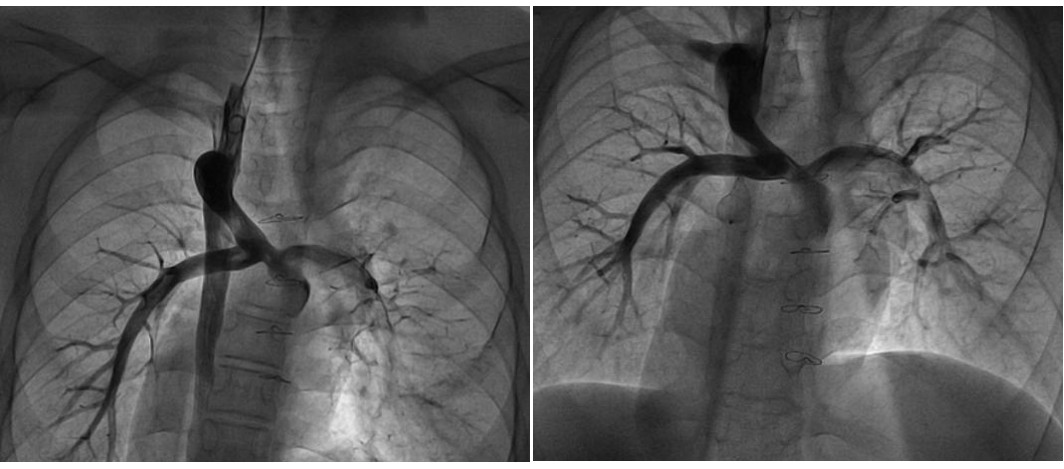

**Figure 1  Interventional closure of azygos vein.**

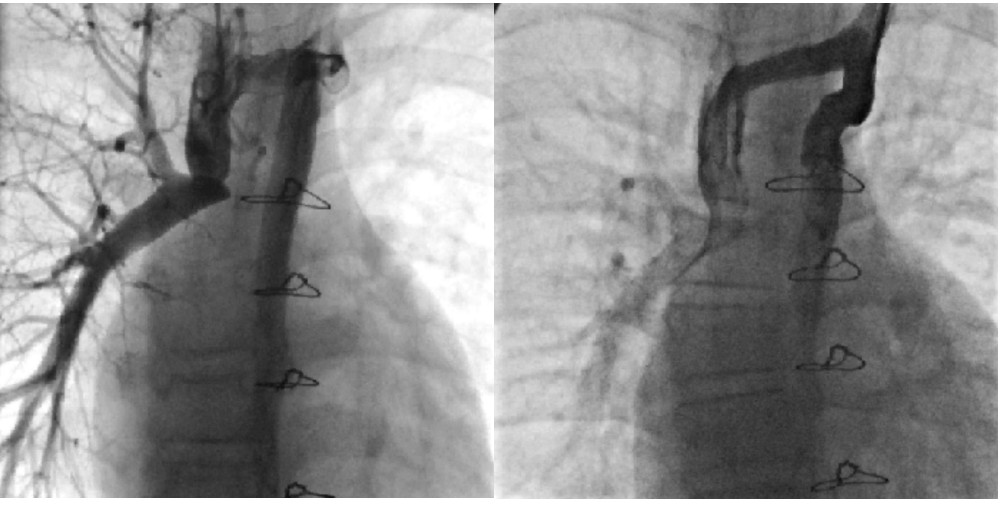

**Figure 2  Interventional occlusion of the hemiazygos vein.**

azygos vein (Fig. 1B). The procedure was uneventful, with no adverse reactions noted in the patient, and a total of 60 mL of contrast medium was used.

(2) The patient underwent transcatheter closure of the hemiazygos vein due to "steal" syndrome. Under local anesthesia, the operator punctured both the left and right jugular veins and inserted a 5F side-hole catheter to measure pressures in the right pulmonary artery, superior vena cava, and hemiazygos vein before and after the closure, as well as to perform hemiazygos vein angiography (Fig. 2A). Subsequently, through the left jugular vein, a 16-mm ASD occluder was successfully implanted into the hemiazygos vein. Angiography confirmed good positioning and shape of the occluder, with no residual shunt (Fig. 2B).

## Follow-up

The patients were followed up regularly in the outpatient clinic or *via* telephone. During these visits, physical examinations were conducted, saturation of pulse oximetry(SpO2) was measured, and electrocardiograms, chest X-rays, and echocardiograms were performed.

## Statistical analysis

All data were processed and analyzed using SPSS(version 27.0) statistical software. For all quantitative data, a normality test was performed. Data that followed a normal or approximately normal distribution are presented as mean ± standard deviation (SD), while data that followed a severely skewed distribution are presented as median (P25, P75) (P25 and P75 are the 25th percentile and 75th percentile respectively). This study employed a paired experimental design with pre-post comparisons. Paired *t*-tests or Wilcoxon signed-rank tests were used for statistical analysis of relevant data before and after the closure. If the differences between pre-closure and post-closure data followed a normal or approximately normal distribution, a paired *t*-test was used. If the differences followed a severely skewed distribution, a Wilcoxon signed-rank test was used. The significance level was set to 5% for a two-sided test. A *p*-value <0.05 indicates statistical significance.

# RESULTS

## Operation results

All azygos or hemiazygos veins were successfully occluded. Among them, 11 cases (85%) achieved complete closure after deployment of the occluder, while two cases (15%) exhibited minor residual shunts. We consider that minor residual shunts are not hemodynamically significant and may spontaneously close over time. Therefore, no intervention is required for these cases, and follow-up observation is sufficient. During the occlusion of the azygos or hemiazygos veins, most patients received ASD occluders, with only two patients receiving patent ductus arteriosus (PDA) occluders. Case 3 had preoperative stenosis at the cavopulmonary anastomosis, which was successfully treated with balloon dilation during the procedure, resolving the stenosis. Case 9 had a pulmonary arteriovenous fistula preoperatively, which was successfully closed with embolization during the procedure. Detailed data regarding surgical outcomes and occluder selection are provided in Table 2. Post-occlusion $S_{FA}O_2$% significantly increased compared to pre-occlusion levels, with a statistically significant difference (86.94 ± 2.63% *vs* 74.98 ± 3.53%, $P < 0.001$, D: 11.96, 95% confidence interval (CI) [9.63–14.29]). Post-occlusion mean pulmonary arterial pressure (mPAP) increased compared to pre-occlusion levels, with a statistically significant difference, but remained within normal ranges (12.08 ± 2.75 mmHg *vs* 10.54 ± 3.28 mmHg, $P = 0.020$, D: 1.54, 95% CI [0.29–2.79]). Post-occlusion diastolic pulmonary artery pressure (dPAP) also increased compared to pre-occlusion levels, with a statistically significant difference, but remained within normal ranges (10.31 ± 3.04 mmHg *vs* 8.46 ± 3.33 mmHg, $P = 0.020$, D: 1.85, 95% CI [0.35–3.34]). The systolic pulmonary artery pressure (sPAP) before and after occlusion (12.69 ± 3.90 mmHg *vs* 14.08 ± 3.30 mmHg, $P = 0.142$, D: 1.38, 95% CI [−0.53–3.30]) and the superior vena cava

**Table 2   This document shows the results of interventional surgery and the choice of occluder.**

| Case | Sex | Age (year) | Vascular access | Occlusion device | Delivery sheath size | Anesthesia | Residual shunt |
|------|-----|-----------|-----------------|------------------|----------------------|------------|----------------|
| 1  | M | 11 | RIJV      | 12 mm ASD occluder     | 5F | LA | no      |
| 2  | M | 15 | RIJV      | 16 mm ASD occluder     | 5F | LA | no      |
| 3  | M | 8  | RIJV      | 12/10 mm PDA occluder  | 5F | GA | trivial |
| 4  | M | 11 | RIJV      | 16 mm ASD occluder     | 5F | LA | trivial |
| 5  | M | 15 | RIJV+LIJV | 16 mm ASD occluder     | 5F | LA | no      |
| 6  | F | 8  | RIJV      | 12 mm ASD occluder     | 5F | GA | no      |
| 7  | M | 9  | RIJV      | 12 mm ASD occluder     | 5F | LA | no      |
| 8  | F | 36 | RIJV      | 16 mm ASD occluder     | 5F | LA | no      |
| 9  | M | 17 | RIJV      | 20 mm ASD occluder     | 5F | LA | no      |
| 10 | F | 17 | RIJV      | 16 mm ASD occluder     | 5F | LA | no      |
| 11 | F | 13 | RIJV      | 12 mm ASD occluder     | 5F | GA | no      |
| 12 | M | 9  | RIJV      | 18/16 mm PDA occluder  | 5F | LA | no      |
| 13 | M | 12 | RIJV      | 16 mm ASD occluder     | 5F | LA | no      |

**Notes.**

M, male; F, female; RIJV, right internal jugular vein; LIJV, left internal jugular vein; GA, general anesthesia; LA, local anesthesia; ASD: atrial septal defect; PDA, patent ductus arteriosus.

pressure (SVCP) before and after occlusion ($11.08 \pm 3.62$ mmHg $vs$ $12.31 \pm 3.25$ mmHg, $P = 0.059$, D: 1.23, 95% CI [$-0.05$–2.52]) showed almost no change, with no statistically significant differences. Other related data before and after occlusion also showed no statistically significant differences. We believe that the possible reason for this is that most post-occlusion data were collected immediately or on the second day after surgery, and too short a time may have prevented observing any significant changes. Additionally, the post-occlusion SVCP ($12.31 \pm 3.25$) and mPAP ($12.08 \pm 2.75$) showed almost no pressure gradient, suggesting no stenosis at the cavopulmonary anastomosis. Detailed hemodynamic data including $S_{FA}O2\%$ before and after the procedure are provided in Table 3.

### Evaluation of follow-up

During an average follow-up of $25.69 \pm 9.60$ months (12 to 50 months), cyanosis improved significantly in all patients, and none experienced facial or neck edema or jugular vein distention. SpO2 levels were above 90% in all patients. There was no recanalization of the azygos or hemiazygos veins, and the two patients with minor residual shunts showed no significant changes. Three patients underwent Fontan procedures within six months post-operatively and recovered well.

### DISCUSSION

BGP represents a standard palliative surgical technique utilized in the management of CCHD, particularly for patients with single-ventricle cardiac malformations or those who are unable to withstand biventricular corrective surgery. For patients with high-risk factors for Fontan surgery, BGP can be part of a staged Fontan approach, enhancing surgical success rates, improving cardiac function, and increasing long-term survival. For those who are not

**Table 3  Comparison of the hemodynamic changes in patients before and after interventional surgery.**

| Azygos/ hemiazygos vein | Pre | Post | t/z | P | D and 95% CI |
|---|---|---|---|---|---|
| SBP, mmHg | 106.15 ± 16.98 | 101.00 ± 5.52 | t = −1.19 | 0.258 | −5.15 (−14.61, 4.30) |
| DBP, mmHg | 71.85 ± 14.59 | 69.77 ± 8.05 | t = −0.501 | 0.625 | −2.08 (−11.11, 6.96) |
| CTR | 0.48 ± 0.07 | 0.47 ± 0.05 | z = −0.68 | 0.498 | / |
| $S_{FA}O_2$% | 74.98 ± 3.53 | 86.94 ± 2.63 | t = 11.19 | <0.001 | 11.96 (9.63, 14.29) |
| sPAP, mmHg | 12.69 ± 3.90 | 14.08 ± 3.30 | t = 1.57 | 0.142 | 1.38 (−0.53, 3.30) |
| dPAP, mmHg | 8.46 ± 3.33 | 10.31 ± 3.04 | t = 2.69 | 0.020 | 1.85 (0.35, 3.34) |
| mPAP, mmHg | 10.54 ± 3.28 | 12.08 ± 2.75 | t = 2.69 | 0.020 | 1.54 (0.29, 2.79) |
| SVCP, mmHg | 11.08 ± 3.62 | 12.31 ± 3.25 | t = 2.09 | 0.059 | 1.23 (−0.05, 2.52) |
| AD, mm | 23.15 ± 3.85 | 22.23 ± 3.30 | z = −1.27 | 0.203 | / |
| AS, mm | 29.00 (26.50, 30.50) | 28.00 (26.00, 33.00) | t = 1.26 | 0.231 | 0.62(-0.45,1.68) |
| AA, mm | 27.54 ± 6.06 | 28.46 ± 5.13 | t =1.17 | 0.264 | 0.92 (−0.79, 2.64) |
| LA, mm | 26.85 ± 5.91 | 26.92 ± 7.42 | t = 0.08 | 0.936 | 0.07 (−1.96, 2.12) |
| LA/Aorta | 0.97 ± 0.29 | 1.02 ± 0.30 | t = −0.98 | 0.346 | −0.05 (−0.16, 0.06) |
| LV, mm | 40.62 ± 10.00 | 39.54 ± 8.30 | z = −0.08 | 0.937 | / |
| EF, % | 62.00 (58.00, 64.75) | 60.00 (60.00, 64.45) | z = −0.27 | 0.789 | / |
| MPA, mm | 13.31 ± 2.32 | 14.31 ± 3.15 | t = 1.03 | 0.325 | 1.00 (−1.12, 3.12) |
| LPA, mm | 9.98 ± 2.58 | 10.40 ± 2.81 | t = 0.73 | 0.482 | 0.42 (−0.85, 1.69) |
| RPA, mm | 10.23 ± 3.35 | 10.36 ± 3.21 | t = 0.20 | 0.847 | 0.13 (−1.31, 1.57) |
| PVSV, m/s | 3.90 (3.20, 4.26) | 3.81 (2.52, 3.81) | z = −0.18 | 0.861 | / |
| PVSPG, mmHg | 60.80 (45.10, 72.65) | 49.00 (13.80, 68.70) | z = −0.63 | 0.529 | / |
| MVDV, m/s | 0.60 (0.50, 0.70) | 0.60 (0.50, 0.70) | z = −0.31 | 0.760 | / |
| MVDPG, mmHg | 1.40 (1.00, 2.00) | 1.40 (1.15, 1.95) | z = −0.34 | 0.735 | / |
| AVSV, m/s | 0.99 ± 0.16 | 0.99 ± 0.19 | t = 0.11 | 0.916 | 0.00 (−0.08, 0.08) |
| AVSPG, mmHg | 3.95 ± 1.22 | 4.00 ± 1.55 | t = 0.24 | 0.812 | 0.06 (−0.49, 0.61) |

**Notes.**

The letters t and z denote the test statistics for the paired t-test and the Wilcoxon signed-rank test, respectively.

D, difference value; CI, confidence interval; SBP, systolic blood pressure; DBP, diastolic blood pressure; CTR, cardiothoracic ratio; $S_{FA}O_2$, femoral artery oxygen saturation; sPAP, systolic pulmonary artery pressure; dPAP, diastolic pulmonary artery pressure; mPAP, mean pulmonary artery pressure; SVCP, superior vena cava pressure; AD, Aortic diameter; AS, anteroposterior diameter of the aortic sinus; AA, ascending aorta; LA, anteroposterior diameter of left atrium; LV, left ventricular end-diastolic diameter; EF, ejection fraction; MPA, main pulmonary artery; LPA, left pulmonary artery; RPA, right pulmonary artery; PVSV, Pulmonary valve systolic velocity; PVSPG, Pulmonary valve systolic pressure gradient; MVDV, Mitral valve diastolic velocity; MVDPG, Mitral valve diastolic pressure gradient; AVSV, Aortic valve systolic velocity; AVSPG, Aortic valve systolic pressure gradient.

suitable for Fontan surgery due to various reasons, BGP can serve as a definitive procedure, allowing for long-term survival (*Calvaruso et al., 2008*). Although the efficacy of the BGP has been established, patients may still face issues such as worsening hypoxia and cyanosis postoperatively. Common causes include an increase in oxygen demand as the patient grows and a reduction in the proportion of blood flow from the SVC relative to total cardiac return, leading to decreased effective pulmonary blood flow and ultimately resulting in an oxygen supply deficit. Additionally, venous collaterals and pulmonary arteriovenous fistulas (PAVF) are common causes of worsened cyanosis. The cavopulmonary pathway, which has relatively higher pressure post-BGP, undergoes decompression through venous collateral channels into the lower pressure IVC, right atrium, or pulmonary veins, representing an adaptive process to the postoperative hemodynamics. However, long-term shunting not

only directly reduces pulmonary blood flow and increases cardiac preload but also leads to an increase in unsaturated blood flow entering the right atrium or pulmonary veins, thereby significantly decreasing arterial oxygen saturation and exacerbating the patient's cyanosis. The occurrence of venous collateral openings post-BGP is not uncommon, with an incidence rate of approximately 17%–33%, among which the azygos or hemiazygos vein openings are the most prevalent (*Andrews, Tulloh & Anderson, 2002*; *McElhinney et al., 1997*). Approximately 80% of venous collateral vessels originate from the innominate vein or its junction with the superior vena cava, with about 55% of these collateral vessels crossing the diaphragm to drain into the IVC (*McElhinney et al., 1997*). The timing of the appearance of venous collateral vessels varies; there have been reports of azygos vein dilation occurring within days postoperatively due to acute thrombosis in the cavopulmonary pathway (*Magee et al., 1998*), as well as reports of azygos vein dilation appearing as late as 16 years after surgery (*Bagul, Singh & Kerkar, 2016*). However, venous collateral vessels are generally detectable between 1 to 3 years following BGP (*Andrews, Tulloh & Anderson, 2002*; *McElhinney et al., 1997*; *Magee et al., 1998*). An increase in pressure within the cavopulmonary pathway is the main cause of venous collateral formation post-BGP, which can be due to thrombosis in the cavopulmonary pathway, stenosis at the cavopulmonary anastomosis, poor development of the pulmonary artery, or insufficient banding of the pulmonary artery (*McElhinney et al., 1997*). *Sawada et al. (2007)* reported a case of a pediatric patient post-BGP who developed obstructive sleep apnea-hypopnea syndrome due to tonsillar hypertrophy, which subsequently led to pulmonary hypertension and the formation of venous collateral vessels. The pressure gradient created between the SVC and the IVC following BGP inherently favors the development of venous collateral vessels. Studies have indicated that the pressure difference between the SVC and the right atrium is an independent predictor of collateral vessel formation (*Magee et al., 1998*). The postoperative opening of potential venous channels that were not patent pre-BGP represents another possible mechanism for the formation of venous collateral vessels, including various venous vessels that become occluded after birth, and even the left SVC, which can reopen following the surgery (*Filippini et al., 1998*). Additionally, studies have shown that vascular endothelial growth factor (VEGF) levels are significantly elevated in children with CCHD, and this increased VEGF may be associated with the formation of venous collateral vessels post-BGP (*Ootaki et al., 2003*). The development of PAVF following BGP is not uncommon, particularly in patients with left atrial isomerism (polysplenia syndrome) who have undergone Kawashima surgery (which is essentially the same as BGP), with an incidence rate as high as 32% and increasing annually (*Kopf et al., 1990*; *Lively-Endicott & Lara, 2018*). Similar to venous collateral vessels post-BGP, PAVF essentially represents an extracardiac right-to-left shunt. Blood within the pulmonary artery bypasses the alveoli and directly flows into the pulmonary veins, creating a "functional arteriovenous shunt", leading to a ventilation/perfusion (V/Q) mismatch and exacerbating the patient's cyanosis. If the PAVF is large, the risk of secondary erythrocytosis and paradoxical embolism also increases (*Van De Bruaene & Budts, 2023*). It is generally believed that the formation of PAVF after BGP is related to the loss of hepatic venous blood flow (containing certain factors that inhibit angiogenesis) perfusion to the pulmonary

circulation following ligation of the main pulmonary artery during surgery (*Van De Bruaene & Budts, 2023*). Since introducing hepatic venous blood flow into the pulmonary circulation can prevent or reverse most PAVF (*Kavarana et al., 2014*; *Juaneda et al., 2018*), our hospital typically maintains the patency of the main pulmonary artery during BGP to reduce the incidence of PAVF, increase arterial oxygen saturation, and promote pulmonary artery development. To avoid excessive forward flow from the ventricle to the pulmonary artery that could impede superior vena cava blood entry into the pulmonary circulation, increase ventricular volume load, and exacerbate atrioventricular valve insufficiency, we employ "pulmonary artery banding" to control the forward blood flow through the pulmonary artery, maintaining it at an appropriate level.

Due to the collateral vessels between the superior and inferior vena cava after BGP, such as the azygos and hemiazygos veins, which naturally resolve their shunting function after Fontan surgery, there is no consensus on whether aggressive management is necessary. We believe that if the azygos or hemiazygos vein is significantly dilated with substantial shunting, leading to a marked decrease in arterial oxygen saturation (as in the cases of our group), and if the patient requires a delayed Fontan procedure or does not meet the indications for Fontan surgery for various reasons, with BGP intended as the final surgical intervention, eliminating the azygos or hemiazygos vein shunt can increase pulmonary blood flow, improve arterial oxygen saturation, reduce cardiac workload, and enhance cardiac function. Historically, the elimination of azygos or hemiazygos vein shunts has been primarily surgical, allowing the surgeon to ligate the azygos or hemiazygos vein directly under visualization and address any associated lesions, potentially offering a more comprehensive treatment. However, surgical intervention often requires thoracotomy, cardiopulmonary bypass, and general anesthesia, which not only entails significant trauma and high risk (such as infection, massive hemorrhage, respiratory complications, *etc.*), but also slow postoperative recovery and noticeable scarring. Additionally, the tissue adhesions caused by surgery can increase the risk of future Fontan procedures. Although there have been reports of thoracoscopic minimally invasive ligation of the azygos vein (*Payne, Bensky & Hines, 2000*), this approach has limitations such as restricted surgical visibility, high technical requirements, and expensive equipment. In recent years, with the rapid development of interventional radiology, the interventional occlusion of the azygos or hemiazygos vein has emerged as an ideal alternative to surgery due to its minimally invasive nature, efficiency, reduced complications, shorter hospital stay, cost savings, and ability to reach areas that are difficult to access through open surgery. Nevertheless, current research on interventional treatment of the azygos or hemiazygos vein is limited, with the majority of cases being reported as individual case studies (*Croti et al., 2009*). *Khajali et al. (2021)* reported the first case of closing the azygos vein using a VSD occluder. The patient, who had undergone BGP for CCHD, developed progressive cyanosis and dyspnea seven years later due to the azygos vein not being ligated during surgery, with an arterial oxygen saturation of 76%. Subsequently, the patient underwent interventional treatment, where the operator implanted a VSD occluder into the azygos vein, resulting in an immediate increase in arterial oxygen saturation to 83% and a significant improvement in hypoxia symptoms. *Maddali et al. (2024)* reported a case of a 5-year-old child who

experienced progressive dyspnea and cyanosis following BGP. Cardiac catheterization revealed that the azygos vein, which had been ligated during BGP, had recanalized, forming an extracardiac right-to-left shunt and causing hypoxic symptoms. The operator then successfully closed the azygos vein by implanting a vascular plug *via* catheter, leading to a stable postoperative condition in the child. A study by *Lu et al. (2014)* included nine patients with CCHD who underwent transcatheter closure of azygos or hemiazygos vein "steal" after BGP, aiming to evaluate the safety and efficacy of interventional therapy. All patients were successfully occluded. Post-occlusion, 81.8% of vessels were completely occluded, and 18.2% had a slight residual shunt. $S_{FA}O_2$% increased from 81% to 88%, while SVCP and mPAP remained largely unchanged. No serious adverse events occurred during the follow-up period. A retrospective study by *Dai et al. (2020)* included 24 patients who underwent transcatheter closure of azygos or hemiazygos vein "steal" after BGP surgery, representing the largest sample size to date for evaluating the safety and efficacy of interventional occlusion of these vessels. All patients were successfully occluded, with SpO2 increasing from 78% to 85% ($P < 0.05$) after occlusion, and there was no significant change in SVCP before and after occlusion (13.94 mmHg *vs* 14.22 mmHg, $P > 0.05$). Following up for 2 years, none of the patients developed limb or facial edema, abdominal wall varicosities, and there was no significant decrease in SpO2. Our study included 13 patients who underwent interventional occlusion of azygos or hemiazygos vein "steal" after BGP surgery. All azygos or hemiazygos veins were successfully occluded, with 85% of vessels completely occluded and 15% showing a slight residual shunt post-occlusion. The $S_{FA}O_2$% significantly increased after occlusion compared to before, with a statistically significant difference ($86.94 \pm 2.63$% *vs* $74.98 \pm 3.53$%, $P < 0.001$, D: 11.96, 95% CI [9.63–14.29]). The mPAP increased significantly after occlusion compared to before, but remained within the normal range, with a statistically significant difference ($12.08 \pm 2.75$ mmHg *vs* $10.54 \pm 3.28$ mmHg, $P = 0.020$, D: 1.54, 95% CI [0.29–2.79]). There was no significant change in SVCP before and after occlusion, with no statistically significant difference ($11.08 \pm 3.62$ mmHg *vs* $12.31 \pm 3.25$ mmHg, $P = 0.059$, D: 1.23, 95% CI [−0.05–2.52]). During an average follow-up period of $25.69 \pm 9.60$ months (12 to 50 months), all patients experienced significant improvement in cyanosis. There were no instances of facial or neck edema or jugular venous distension, and all patients maintained a SpO2 of 90% or above. No recanalization of the azygos or hemiazygos veins was observed in any patient, and the two patients with minor residual shunting showed no significant changes. To our knowledge, our study represents one of the largest sample sizes in research on the interventional treatment of azygos or hemiazygos vein "stealing" after BGP, and it is the first to provide detailed hemodynamic changes before and after the occlusion of these veins. These studies collectively indicate that the interventional treatment of azygos or hemiazygos vein "stealing" post-BGP is a safe and effective approach, with high technical success rates and favorable short- and intermediate-term prognoses, thereby avoiding the physical and psychological trauma associated with secondary thoracotomy. During the interventional occlusion of the azygos or hemiazygos veins, coexisting lesions such as cavopulmonary anastomosis stenosis, pulmonary artery stenosis, and PAVF can be addressed simultaneously. Balloon dilation is a safe and effective technique for relieving

anastomotic or vascular stenosis, and in cases where necessary, a stent can be implanted to prevent recurrent stenosis (*Matoq & Radtke, 2020*). We recommend interventional embolization for solitary cystic PAVF post-BGP, which has a proven efficacy and a low incidence of complications. However, the management of multiple or diffuse PAVF in both lungs can be quite challenging. For severe cases with multiple bilateral PAVF, larger lesions may be selectively occluded to improve arterial oxygen saturation and alleviate symptoms of hypoxia.

The selection of an appropriate occluder for the azygos or hemiazygos veins based on the severity of shunting and the size of the vessel lumen is crucial for interventional therapy. It is essential to achieve complete occlusion of the vessel without interfering with adjacent structures. Typically, for mild shunting of the azygos or hemiazygos veins (≤25% of the diameter of the innominate vein), no intervention is generally necessary and follow-up observation is sufficient; for moderate shunting with mild to moderate dilation of the azygos or hemiazygos veins (≤50% of the diameter of the innominate vein), coils are recommended; for severe shunting with severe dilation of the azygos or hemiazygos veins (>50% of the diameter of the innominate vein), ASD or PDA occluders are suggested (*Lu et al., 2014*). We typically opt for ASD occluders because the walls of veins are thinner than those of arteries, and this type of occluder exerts less tension after closure, potentially reducing related complications. Patients may experience transient chest pain postoperatively, which could be due to the tension exerted by the occluder or the occluder compressing certain sensory nerve fibers (*Lu et al., 2014*). This phenomenon was not observed in the patients in our study, which may be related to the fact that the majority of patients (85%) chose ASD occluders.

## Limitation

This study demonstrates that interventional treatment for azygos or hemiazygos vein "stealing" post-BGP is safe and effective. However, this study also has certain limitations. Firstly, it is a single-center, small sample size, retrospective study, which inherently makes it susceptible to certain biases. The conclusions drawn from such studies can only provide preliminary evidence and support for clinical practice, and further validation is required through multicenter, large-scale, prospective studies in the future. Secondly, although this study is one of the largest sample sizes assessing the safety and efficacy of interventional treatment for azygos or hemiazygos vein "stealing" post-BGP, with an average follow-up of 25.69 ± 9.60 months (12–50 months), the short-term and intermediate clinical outcomes are encouraging, the long-term outcomes remain unclear. Future studies with extended follow-up durations are warranted to validate these findings. Lastly, although this study is the first to provide detailed hemodynamic changes before and after the occlusion of the azygos or hemiazygos veins, many of the hemodynamic changes post-surgery did not reach statistical significance. This is because the majority of our post-occlusion data were collected immediately after surgery or the following day, and the short timeframe did not allow for the observation of significant effects.

## CONCLUSIONS

Interventional treatment of azygos or hemiazygos vein "stealing" post-BGP in patients with CCHD is a safe and effective approach. It can increase pulmonary artery blood flow, elevate arterial oxygen saturation, alleviate cardiac load, and enhance cardiac function. Moreover, it has a high technical success rate and favorable short- and intermediate-term prognoses, avoiding the physical and psychological trauma associated with secondary thoracotomy.

### Funding

This study was funded by the Beijing Municipal Science and Technology Commission "Capital Clinical Characteristic Diagnosis and Treatment Technology Research and Transformation Application" Special Fund (Z201100005520075), Central High-Level Hospital Clinical Research Operating Funds (2022-GSP-GG-18) and China Academy of Medical Sciences Central-Level Public Welfare Research Institute Basic Scientific Research Operating Funds (2022-RW320-09). The funders had no role in study design, data collection and analysis, decision to publish, or preparation of the manuscript.

### Grant Disclosures

The following grant information was disclosed by the authors:
Beijing Municipal Science and Technology Commission: Z201100005520075.
Central High-Level Hospital Clinical Research Operating Funds: 2022-GSP-GG-18.
China Academy of Medical Sciences Central-Level Public Welfare Research Institute Basic Scientific Research Operating Funds: 2022-RW320-09.

### Competing Interests

The authors declare there are no competing interests.

### Author Contributions

- Zhengwei Li conceived and designed the experiments, performed the experiments, analyzed the data, prepared figures and/or tables, and approved the final draft.
- Luxi Guan performed the experiments, analyzed the data, prepared figures and/or tables, and approved the final draft.
- Dong Luo analyzed the data, prepared figures and/or tables, and approved the final draft.
- Meijun Liu analyzed the data, prepared figures and/or tables, and approved the final draft.
- Haibo Hu conceived and designed the experiments, authored or reviewed drafts of the article, and approved the final draft.
- Xiangbin Pan conceived and designed the experiments, authored or reviewed drafts of the article, and approved the final draft.

### Human Ethics

The following information was supplied relating to ethical approvals (i.e., approving body and any reference numbers):

The study has been approved by the ethics committee of Fuwai Hospital (2021-1551).

## Data Availability

The raw measurements are available in the Supplemental File.

## Supplemental Information

Supplemental information for this article can be found online at http://dx.doi.org/10.7717/peerj.20022#supplemental-information.

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
