# Peer review of "Interventional treatment for azygos vein steal syndrome after bidirectional Glenn procedure in cyanotic congenital heart disease: a retrospective study"

_PeerJ, doi:10.7717/peerj.20022_

## Round 0.1 · original submission · Minor Revisions

· Academic Editor

Minor Revisions

Please respond to the comments from the reviewers in an appropriate revision

Reviewer 1 ·

Basic reporting

The authors provide a thoughtful literature review and case series of 13 patients in 1 clinical center undergoing transcatheter closure of azygous vein for "steal" syndrome after bidirectional glenn for cyanotic heart disease. The introduction and discussion are well written, with an extensive literature review and thoughtful commentary on physiologic adaptations in the setting of azygous venous steal post bidirectional glenn.

Experimental design

This is a retrospective analysis of 13 cases over a 12 year period, outlined as a retrospective study, but the design is more in keeping with a case series from one surgical center and should probably be reframed as such. The authors mainly seek to demonstrate safety and feasibility of percutaneous closure in this short series. The main limitation of this is the evolution of technology and percutaneous methodology over 12 years and therefore the heterogeneity in these cases, which is why framing this a case series is more scientifically sound than a retrospective study with only 13 cases over 12 years. Otherwise the methods and results are thoughtfully written and sound, easy to follow and outline the objectives of the study in terms of feasibility and safety.

Validity of the findings

The main limitations of this paper is twofold. First is the heterogeneity in a small cohort of patients that are presented to a single center over a 12 year period, with variable initial cyanotic heart disease lesion, evolving surgical expertise and technology in precutaneous interventions. The best way to address these limitations is to outline this as a case series of one surgical center rather than a retrospective cohort study. Second is the very short follow up period of 1 year post closure. Given that the pathophysiology of steal is collateralization which takes time to develop, it limits the implications that can be drawn from this case series to only report data 1 year post procedure which should be acknowledged in the limitations. Apart from these 2 limitations, the findings are reasonable and the author provide a very nicely laid out commentary on the physiology of this phenomenon, the management challenges, and pitfalls.

Reviewer 2 ·

Basic reporting

The manuscript is clearly written, although there are some minor comments below:
o Line 133 “The patient was followed up”, should be “the patients were followed up”
o Line 147-148, “The significance level was set at a two-tailed p-value of 0.05, with p < 0.05 indicating statistically significant differences”, suggest to rephrase it to “The significance level was set to 5% for a two-sided test. A p-value <0.05 indicates statistical significance.”
o Line 370: “Conclusions” instead of “conclusions”
o There should be a comma to separate lower and upper bound of 95% confidence interval in table 3

Experimental design

o It would be good to clarify the time when pre- and post-occlusion measurements of SFAO2% and hemodynamic data were taken. Are there any summary statistics such as mean and standard deviation showing how long these measurements were collected before and after occlusion?
o It was noted that post-occlusion measurements of SFAO2% and hemodynamic data were collected shortly after the surgery. Were any of these measurements collected during post-surgery follow-up period?
o During the follow-up period, is there an intent to evaluate the ECG (electrocardiograms) parameters. And how about the dynamic change in SpO2 levels?
o Statistical analysis: Has the one-sided statistical test been considered, instead of two-sided test?

Validity of the findings

o In table 3: clarify what t and z mean (are they test statistics for paired sample t test and signed rank test?)
o Line 142, need to clarify P25 and P75 are the 25th percentile and 75th percentile respectively
o It was noted in the statistical analysis section that data are presented as median and 25th and 75th percentile and Wilcoxon signed-rank test is used if the distribution is severely skewed. From table 3, there are variables presented as median and 25th and 75th percentiles, for example, AS (anteroposterior diameter of the aortic sinus). Was this variable tested by signed-rank test? But why is the test statistic t the same as that of variable that is tested by paired t-test (for example variable AA)?

---

## Round 0.2 · accepted · Accept

· Academic Editor

Accept

We acknowledge that the revisions you submitted in response to the reviewers' comments were appropriate and thoroughly addressed all concerns raised during the peer review process. Based on the scientific merit of your study and the quality of your responses, we are confident that your manuscript meets the standards of PeerJ and is suitable for publication.

Reviewer 2 ·

Basic reporting

The authors have addressed all previous comments.

Experimental design

-

Validity of the findings

-